# Change of Ruminoreticular Temperature and Body Activity before and after Parturition in Hanwoo (*Bos taurus coreanae*) Cows

**DOI:** 10.3390/s21237892

**Published:** 2021-11-26

**Authors:** Daehyun Kim, Jaejung Ha, Woo-Sung Kwon, Joonho Moon, Gyeong-Min Gim, Junkoo Yi

**Affiliations:** 1Livestock Research Institute, 186 Daeryongsan-ro, Anjeong-Myeon, Yeongju 36052, Gyeongsangbuk-do, Korea; chunja2411@korea.kr (D.K.); hjjggo@korea.kr (J.H.); 2Department of Animal Science and Biotechnology, Kyungpook National University, Sangju 41566, Korea; wskwon@knu.ac.kr; 3Lartbio Co., Ltd., 12 Floor, 234 Teheran-ro, Gangnam-gu, Seoul 06221, Korea; kuma618@gmail.com; 4Laboratory of Theriogenology and Biotechnology, Department of Veterinary Clinical Sciences, College of Veterinary Medicine, Seoul National University, 1 Gwanak-ro, Gwanak-gu, Seoul 08826, Korea; tty4447@naver.com

**Keywords:** parturition, bio-capsule sensor, ruminoreticular temperature, body activity, Hanwoo

## Abstract

How do body temperature and activity change before and after parturition in pregnant cows? Changes in body temperature such as ruminal, rectal, and vaginal temperature during the parturition have been reported, but there are no results of the simultaneous observation of body temperature and activity. The aim of this study was to simultaneously confirm changes in the ruminoreticular temperature and body activity before and after parturition using the ruminoreticular bio-capsule sensor every 1 h. The 55 pregnant cows were used for the experiment, the ruminoreticular bio-capsule sensor was inserted and stabilized, and the ruminoreticular temperature and body activity were measured. The ruminoreticular temperature was lower by 0.5° from −24 h to −3 h in parturition compared to 48 h before parturition and then recovered again after parturition. Body activity increased temporarily at the time of parturition and 12 h after parturition. Therefore, the ruminoreticular temperature and body activity before and after parturition was simultaneously confirmed in pregnant cows.

## 1. Introduction

In the case of cows, since they are ruminants, their health status, body activity, and ruminoreticular temperature are very closely related [1]. Recently, a bio-capsule sensor-based ICT system (bolus system) has developed a technology that can detect changes in ruminoreticular temperature in real-time through the ICT equipment by inserting and settling it in the rumen of cows. This equipment has been used to conduct various research investigations such as investigating the changes in the concentration of milk production of cows, ruminoreticular pH, and feed intake rate etc. [1,2].

According to the results of comparing the rectal temperature from 5 to 12 days after artificial insemination, the temperature of the pregnant cow (39.09 ± 0.22 °C) is 0.46 °C higher than the non-pregnant cow (38.63 ± 0.14 °C) [3]. There are research reports that the vaginal temperature of pregnant dairy cattle showed the same changes depending on the pregnancy period, with an average temperature of 38.63 ± 0.14 °C on 67 days of pregnancy and 39.01 ± 0.03 °C on 147 days of pregnancy [4].

As a result of measuring the vaginal temperature at 180 days of pregnancy, there are research reports that the pregnant cow maintains a relatively high temperature compared to the non-pregnant cow: in the case of vaginal temperature, the temperature pf the pregnant cow was 39.10 ± 0.40 °C and the non-pregnant cow 38.80 ± 0.30 °C, and in the case of rectal temperature, the pregnant cow was 38.70 ± 0.30 °C and the non-pregnant cow 38.50 ± 0.50 °C [5].

Changes in ruminoreticular temperature during the pregnancy period of Hanwoo cows seen during previous studies conducted by this research team, gradually decreased at about 40 to 160 days as of the date of artificial insemination from 40 days compared to non-pregnant cows. The results differed the most at 100 days of pregnancy, and then maintained a temperature similar to non-pregnant cows at 180 to 190 days of pregnancy [6]. From 190 days of pregnancy to 280 days, just before parturition, this study has shown that the body temperature was clearly increased compared to the non-pregnant cows [6]. In addition, as a result of statistically analyzing the ruminoreticular mean temperature of pregnant cows in a total of four phases, there was a significant difference in mean temperature in each phase: 38.68 ± 0.01 °C from 80–100 days of pregnancy, 38.78 ± 0.02 °C from 145–165 days, 38.99 ± 0.45 °C from 200–220 days, and 39.14 ± 0.38 °C from 250–270 days [6].

As a result of measuring the vaginal temperature at 180 days of pregnancy, there are research reports that the pregnant cow maintains a relatively high temperature compared to the non-pregnant cow: in the case of vaginal temperature, the pregnant cow was 39.10 ± 0.4 °C and the non-pregnant cow 38.80 ± 0.3 °C. In the case of rectal temperature, the pregnant cow was 38.70 ± 0.3 °C and the non-pregnant cow 38.50 ± 0.5 °C.

In addition, the to the ruminoreticular, rectal, and vaginal temperatures before and after parturition, the research report found that the temperature decreased more than usual compared to the temperature that was reported right before parturition [3,4,7]. In particular, in the case of ruminoreticular temperature, it was reported to have been maintained at an average temperature of 38.94 ± 0.05 °C during the 2 to 7 days before parturition, and then decreased by roughly 0.4 °C to 38.55 ± 0.05 °C just before parturition [7]. As a result of analyzing the activity before delivery through electronic data loggers attached to the leg, there is a study report that confirms that the temperatures of the heifers were relatively higher than that of dairy cows [8].

## 2. Case Presentation

### 2.1. Animals and Management

The animals utilized in this study were 55 Hanwoo cows, scheduled for parturition, with a Bolus system inserted, and bred at the National Livestock Research Institute in Gyeongsangbuk-do. The age of the over 260 days pregnant cow (*n* = 31) was 52.7 ± 4.6 months of age, and the age of the cow post-delivery (*n* = 24) was 57.7 ± 5.15 months. This study organized a 24 h standby team for parturition and investigated the calving date, time, gender, and birth weight, etc. The animal experiment was conducted after obtaining approval by the Institutional Animal Care and Use Committee and the National Livestock Research Institute in Gyeongsangbuk-do (Approval number: #106). All cows were fed in accordance with the Korean Feeding Standard for Hanwoo cows and raised in a sufficient space where a stanchion was installed.

### 2.2. Measurement of Ruminoreticular Temperature and Body Activity

Bio-capsule sensors (LiveCare, ulikeKorea, Seoul, Korea) were settled in the reticulorumen of all Hanwoo cows used in the study through oral administration with an adaptation period of at least one month. This study aimed to measure the body temperature at intervals of 1 h through a biosensor inserted in the reticulorumen during the last month of pregnancy, and the collected ruminoreticular temperatures were used for analysis. The sensor is 125 mm long, 36 mm in diameter, and weighs 200 g with a battery. The method of collecting ruminoreticular temperatures through biosensors can be found in Kim and Choi et al. [9,10]. Body activity (V) is expressed as the following using an indwelling 3-axis (X, Y and Z) accelerometer. Detailed information has been reported [10].


$$Body\;activity\;=\;\sqrt{X^{2}\;+\;Y^{2}\;+\;Z^{2}}$$


### 2.3. Pregnancy Test

A pregnancy test was performed through via transrectal ultrasonography (HONDA HS-101V, HONDA Co., Ltd., Tokyo, Japan) 15 days before the experiment by experienced veterinarians and technicians.

### 2.4. Statistical Analysis

For statistical analysis, this study utilized the GraphPad PRISM (version: 8.1.0.) program. The ruminoreticular temperature and body activity before and after parturition were analyzed using two-way ANOVA (Sidak’s multiple comparison test), and the level of significance was *p* < 0.01.

### 2.5. Experimental Results

Among the experimental animals, the cows that have recently delivered are referred to as “parturition”, and the cows over 260 days of pregnancy are referred to as “pregnancy (>260 d)” (Figure 1).

In order to observe changes in the ruminoreticular temperature and body activity before and after parturition of Hanwoo cows, this study investigated the actual time of parturition, sex, and birth weight, etc. as shown in Table 1.

Compared to “Pregnant (>260 d)”, “Parturition” showed a specific ruminoreticular temperature change before and after parturition (Figure 1A). As a result of the comparative analysis of the ruminoreticular temperatures before and after parturition, compared to 38.98 ± 0.06 °C at −48 h before parturition, it significantly decreased to 38.49 ± 0.04 °C −24 h to −3 h before parturition (Figure 1A). The ruminoreticular temperature of the “Parturition” group maintained a reduced temperature compared to the “Pregnant (>260 d)” group 16 h after parturition (Figure 1A).

Body activity showed no significant difference between groups; “Pregnant (>260 d)” and “Parturition” for 48 h before delivery (Figure 1B). However, compared to “Parturition”, the body activity of “Pregnant (>260 d)” temporarily increased at the time of parturition (1247.10 ± 11.48 V) and 12 h after parturition (1245.87 ± 15.83 V) (Figure 1B).

## 3. Discussion

Although each research finding of body temperature and body activity before and after parturition of cows has been reported, it is difficult to predict accurate parturition by using a biosensor because it only briefly describes the changes in body temperature and body activity. There is a research report that observed a rapid increase in body activity, maintaining a low temperature in comparison with normal body temperature immediately before parturition [7,8]. However, no research findings have been reported regarding concurrent ruminoreticular temperature and body activity every 1 h.

According to previous studies by this research team, for non-pregnant cows, the ruminoreticular temperature was 38.72 ± 0.08 °C, but the temperature was relatively high at 39.14 ± 0.38 °C on 250–270 days of pregnancy [6]. This study was able to identify a specific pattern before and after parturition, considering ruminoreticular body temperature and body activity by utilizing the bio-capsule sensors.

Similar to the findings published by Cooper Prado et al., there was a low temperature between −24 and −3 h, which is seen as a typical physiological feature. However, more detailed changes in ruminoreticular temperature were confirmed every hour compared to Cooper Prado’s results [7,11]. Therefore, based on this result, it is highly valuable as a criterion for predicting of parturition time based on ruminoreticular temperature [2]. Titler et al. reported that feed and water intake are also related to the ruminoreticular temperature. Therefore, there is a well-known correlation between body temperature and progesterone concentration, and it is estimated that a low temperature phenomenon occurred before delivery due to a temporary reduction in feed intake due to stress or delivery pain [2,7].

The temporary increase in body activity confirmed at the time of delivery and 12 h after parturition relates to the normal movement required to take care of the fetus after parturition. Since the same pattern was confirmed in research results such as Titler et al., it is presumed to be the result of labor at the time of parturition and movement to take care of the calf after parturition.

## 4. Conclusions

In summary, detailed data on ruminoreticular temperature and body activity in cows every 1 h before and after parturition are very important for a more precise prediction of parturition time. Accurately predicting parturition time is highly related to the survival rate of the fetus. Therefore, the results of this study can be used as basic data for improving the accuracy of artificial intelligence systems predicting parturition.

## Figures and Tables

**Figure 1 sensors-21-07892-f001:**
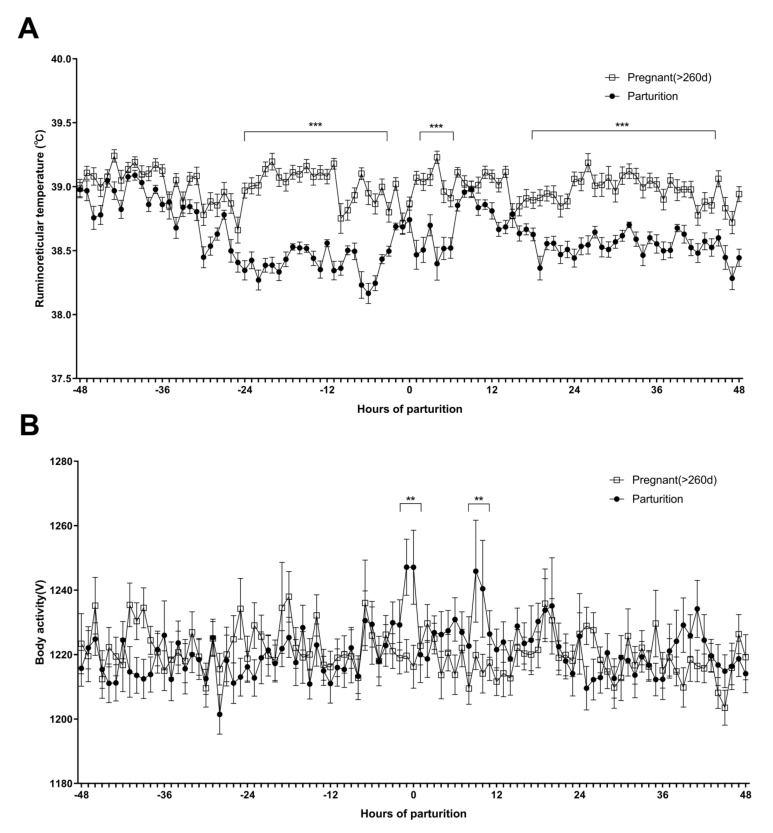
Change of ruminoreticular temperature and body activity before and after parturition **i**n Hanwoo cows (*n* = 55). The “Pregnant (>260 d)” group is 260 days or more pregnant, and the “parturition” group mean the recent delivery of the cow. 0 is the hour of parturition. (**A**) Ruminoreticular temperature of the “Pregnant (>260 d)” and “Parturition” group. The black line connected by “□” represents the mean of the “Pregnant (>260 d)” group. The black line connected by “●” represents the mean of the “Parturition” group. (**B**) Body activity of “Pregnant (>260 d)” and “Parturition” group. The black line connected by “□” represents the mean of the “Pregnant (>260 d)” group. The black line connected by “●” represents the mean of the “Parturition” group. All results are presented as mean ± SEM. (***) significantly different (*p* < 0.001). (**) significantly different (*p* < 0.01).

**Table 1 sensors-21-07892-t001:** Information of parturition and offspring in experiment group (*n* = 24).

Cow ID	Day of Parturition (Year-Month-Day)	Time of Parturition (Hour:Minute)	Sex	Birth Weight (kg)
#13-81	2019-02-05	17:10	Female	20
#16-18	2019-02-22	18:10	Female	27
#16-25	2019-04-01	22:00	Male	30
#16-26	2019-03-01	14:30	Female	30
#16-4	2019-03-06	18:30	Male	27
#17-28	2019-02-27	15:20	Male	24
#17-29	2019-02-24	14:29	Female	22
#17-3	2019-03-07	01:10	Female	24
#17-36	2019-02-26	14:30	Male	28
#17-44	2019-03-03	07:00	Male	27
#19-02	2019-02-08	23:00	Male	30
#19-03	2019-02-09	18:45	Male	28
#19-06	2019-02-14	20:30	Female	38
#19-08	2019-02-15	10:30	Male	28
#19-09	2019-02-15	17:50	Female	31
#19-13	2019-02-17	22:00	Male	19
#19-19	2019-02-20	12:20	Male	32
#19-21	2019-02-21	15:30	Male	30
#19-26	2019-02-25	02:30	Male	28
#19-30	2019-02-27	22:40	Male	34
#391	2019-03-01	11:40	Female	24
#411	2019-03-08	04:20	Female	34
#431	2019-03-17	12:30	Male	39
#432	2019-03-21	01:10	Female	31

## Data Availability

Not applicable.

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
