# Peer review of "Change of Ruminoreticular Temperature and Body Activity before and after Parturition in Hanwoo (Bos taurus coreanae) Cows"

_sensors, 2021, doi:10.3390/s21237892_

Round 1

Reviewer 1 Report

The authors have addressed my main concerns with this paper, so I am now happy to recommend publication.

Reviewer 2 Report

The article has been improved by the authors and I appreciate it, but same as before I would strongly recommend the paper to be prove-read by a  vet/vet-technician or similar, as there are some inaccuracies and misinterpretations of physiological factors, which form my point of view need to be corrected before publication.

 Line 64 – 66: sentence is unclear, what temperature decreased? Compared with which temperature before parturition?

Line 69: what is ruminoreticular body activity? In cows there is either ruminoreticular activity or body activity (= limbs movements etc.) please check, specify and correct throughout text and Line 70: body acitivity increased before parturition or ruminoreticular activity – this is important biological difference please specify

Line 76: the months of age?

Line 88: during one month of pregnancy? Was it the last month of pregnancy or any month during cows’ pregnancy – please specify

Lines 105 -107: so, Parturition group were animals > 260 d of pregnancy whereas Pregnant animals ….? It is unclear please specify

      Line 147: the pre-parturient temperature drop is not a phenomenon, it is a typical physiological finding in cows, please refer to for example: DOI: 10.3168/jds.2011-4484 and re-write the sentence.

      Line 152-154: there is well-known correlation between body temperature and progesterone concentration, progesterone is the hormone which maintains pregnancy and which serum level drops before parturition, which affects body temperature, leading to the temperature drop. Please refer to: M. Sakatani: Reproductive Management by the Continuous Body Temperature Measurement in Cattle: Focusing on the Reproductive Hormonal Change    https://ap.fftc.org.tw/article/1620

Or here: DOI:https://doi.org/10.3168/jds.2011-4484

Lines 160-162: authors observed temperature around parturition so why commenting here on estrous observations?

Author Response

Thank you for your interest and peer review in our study.

Round 2

Reviewer 2 Report

Authors addressed all my comments and suggestions. 

This manuscript is a resubmission of an earlier submission. The following is a list of the peer review reports and author responses from that submission.

Round 1

Reviewer 1 Report

The paper is of interest in determining methods to monitor and manage physiological function e.g. parturition in cattle and other livestock.  However I have found numerous apparent issues with the way the results are presented and described. Specifically:

  1. It is reported in the introduction / M&M that the temperature recordings in the R_R are logged at 10-minute intervals.  The data in Fig 1 are recorded as mean +/- SEM.  Presumably then an average of the 10-min values is taken for each cow to give an average 6-hourly value and then these are meaned for the cohort of cows.  is this correct? If so I am not sure that this is the correct statistical approach as it ignores the intra-cow variance during each 6-hour period.  However i do agree that it looks as if there is a significant drop in temperature from about -30hours before parturition.  However what are the asterisked values significantly different from?
  2. In Fig 2, there is no significant change  in body activity, just a hint that there might be  from about -6 hours.
  3. I am troubled by the description of the changes in steroid concentrations around parturition.  These changes are well-characterised in many previous publications.  However the patterns shown in Fig. 3 do not represent the expected changes in my opinion. For example the fall in progesterone at the end of pregnancy is not at all clear from the data presented; neither is there a characteristic rise in E2 levels; furthermore cortisol levels would be expected to rise prior to parturition and they don't in this data.  In the Discussion it is claimed that these hormonal changes are the same as in other studies, but they do not look like that to me.

Thus in conclusion I am not sure what new data this paper shows.  If it shows more detailed changes of the R-R temperature every 10 minutes for a large cohort of cattle around parturition then this is worth publishing but otherwise I am doubtful about its value.

Reviewer 2 Report

I carefully read the submitted manuscript and after overall consideration, the status of the paper is not suitable for publication. The structure of the paper is difficult to follow and the discussion is quite short and not conclusive. The basic concept of the paper might be interesting for publication, but the argumentation within the paper is not straight.

In this paper, the authors try to make a time of birth prediction with the help of a ruminoreticular sensor, which measures the ruminoreticular temperature and movement.

From the beginning, the paper is difficult to read, caused by wrong grammar and many word repetitions like in this study and in addition.  

Here some points in detail: 

Page 1 Line 16: cows

Page 1 Line 22: The sentence doesn’t make sense at all, so 24 cows were used for the experiment? Bos Taurus coreanea is for meat production right? Why did you choose cows at this age (quite old)? Please mentioned which parameter you included in the parity.

Page 1 Line 25: How did you stabilized the capsule?

Page 1 Line 26 to 29: Please overwork the sentence, there are too many “and” and it´s from 7 days before to 3 days after parturition.

Page 1 Line 29 to 31: Overwork this sentence, a study could not find anything out on its own. I think the statement of this sentence is that the body temperature is increasing 30 hours before parturition.

Replace in this study and make your findings clear. Content of progesterone? Do you mean amount?

Reviewer 3 Report

The aim of the presented study, was to evaluate changes in the ruminoreticular temperature and body activity before and after parturition using the ruminoreticular bio-capsule sensor.

As a reviewer I am not expert is sensors devices, but reviewing this work from biological sciences point of view, the paper is quite hard to follow and often very hard to understand.

In my opinion the biological part is very important for the presented research, as the accuracy of such sensor is proved by the biological trial. So the usefulness and suitability of the sensor would be evaluated only if the biological part of the experiment is well designed and performed. I strongly recommend consulting a vet and/or animal breeders to correct flaws of this paper. I would also like to stress that the content is interesting and would be interesting for the readers, but it needs still some work before considering for publication.

Main concerns:

not specified time frame of experiment, when during pregnancy

time of sensor application in relation to pregnancy

not specified time of pregnancy confirmation and by whom done (experienced vet or technician)

hormone analysis – ELISA kit, who performed readings (experienced staff? device?), the readings of fluorescent or chemiluminescent signals tend to be operator dependant, was it compared to standard curve? Does the producer provides data on these tests accuracy compared to other devices for hormonal analysis available on the market?

English language – needs improvement

authors investigated: ‘ruminoreticular body activity’ or ‘ruminoreticular temperature and body activity’????

really, did the authors need to supervise parturition themselves? The breeding Institute staff do not provide such care for the animals? And the breeding Institutes usually collect data regarding calf sex, birth weight, calving date and time. Do the authors really weighted calves themselves?

Sensor – validation of the accuracy of temperature sensor, was it performed?

The  detailed review is to be provided after main revision of the text and addressing the above mentioned main concerns.